# Single-Cell Sequencing of Lung Macrophages and Monocytes Reveals Novel Therapeutic Targets in COPD

**DOI:** 10.3390/cells12242771

**Published:** 2023-12-05

**Authors:** Yushan Hu, Xiaojian Shao, Li Xing, Xuan Li, Geoffrey M. Nonis, Graeme J. Koelwyn, Xuekui Zhang, Don D. Sin

**Affiliations:** 1Department of Mathematics and Statistics, University of Victoria, Victoria, BC V8P 5C2, Canada; yushanhu@uvic.ca; 2Digital Technologies Research Centre, National Research Council Canada, Ottawa, ON K1A 0R6, Canada; xiaojian.shao@nrc-cnrc.gc.ca; 3Department of Mathematics and Statistics, University of Saskatchewan, Saskatoon, SK S7N 5A2, Canada; lix491@mail.usask.ca; 4UBC Centre for Heart Lung Innovation, St. Paul’s Hospital, Vancouver, BC V6Z 1Y6, Canada; annie.li@hli.ubc.ca (X.L.); geoff.nonis@hli.ubc.ca (G.M.N.); graeme.koelwyn@hli.ubc.ca (G.J.K.); don.sin@hli.ubc.ca (D.D.S.); 5Faculty of Health Sciences, Simon Fraser University, Burnaby, BC V5A 1S6, Canada; 6Division of Respirology, Department of Medicine, University of British Columbia, Vancouver, BC V6T 1Z1, Canada

**Keywords:** macrophages, monocytes, COPD, single-cell RNA sequencing, fluticasone, drug discovery

## Abstract

Background: Macrophages and monocytes orchestrate inflammatory processes in the lungs. However, their role in the pathogenesis of chronic obstructive pulmonary disease (COPD), an inflammatory condition, is not well known. Here, we determined the characteristics of these cells in lungs of COPD patients and identified novel therapeutic targets. Methods: We analyzed the RNA sequencing (scRNA-seq) data of explanted human lung tissue from COPD (n = 18) and control (n = 28) lungs and found 16 transcriptionally distinct groups of macrophages and monocytes. We performed pathway and gene enrichment analyses to determine the characteristics of macrophages and monocytes from COPD (versus control) lungs and to identify the therapeutic targets, which were then validated using data from a randomized controlled trial of COPD patients (DISARM). Results: In the alveolar macrophages, 176 genes were differentially expressed (83 up- and 93 downregulated; P_adj_ < 0.05, |log_2_FC| > 0.5) and were enriched in downstream biological processes predicted to cause poor lipid uptake and impaired cell activation, movement, and angiogenesis in COPD versus control lungs. Classical monocytes from COPD lungs harbored a differential gene set predicted to cause the activation, mobilization, and recruitment of cells and a hyperinflammatory response to influenza. In silico, the corticosteroid fluticasone propionate was one of the top compounds predicted to modulate the abnormal transcriptional profiles of these cells. In vivo, a fluticasone–salmeterol combination significantly modulated the gene expression profiles of bronchoalveolar lavage cells of COPD patients (*p* < 0.05). Conclusions: COPD lungs harbor transcriptionally distinct lung macrophages and monocytes, reflective of a dysfunctional and hyperinflammatory state. Inhaled corticosteroids and other compounds can modulate the transcriptomic profile of these cells in patients with COPD.

## 1. Introduction

Chronic obstructive pulmonary disease (COPD) is an inflammatory disorder of the lungs that affects 384 million people and is responsible for over 3 million deaths/year [1], making it the third leading cause of mortality worldwide. COPD is characterized by irreversible limitation of airflow [2]; however, patients often display different morphological and clinical phenotypes, with the most common being chronic bronchitis, emphysema, and small airway remodeling [3,4,5,6].

It is now well established that COPD is an inflammatory disorder [7]; however, its pathogenesis is largely unknown. In the lungs, the most common immune cells are macrophages. The primary functions of these cells are to eliminate inhaled pathogens, recycle surfactants, initiate tissue repair, and to maintain lung homeostasis [8]. In COPD, these cells become dysregulated, demonstrating impaired phagocytosis, reduced efferocytosis, altered secretion of inflammatory mediators, and oxidative stress. These cells also become less sensitive to therapeutics such as corticosteroids [8] in COPD.

Although, traditionally, macrophages have been classified into the M1/M2 phenotypes, it is now recognized that owing to their plasticity and their ability to adapt to their milieu, this classification scheme is overly simplistic and does not reflect the state of macrophages in vivo [9]. The advent of single-cell sequencing (scRNAseq) technology has revolutionized our ability to identify distinct subpopulations of cells which were previously ignored. A previous study using scRNAseq technology showed that in lung tissues of COPD patients, macrophages and monocytes harbored the greatest number of differently expressed genes compared with those from control subjects without COPD among all cell types [9]. However, this study could not fully identify the important subpopulations of macrophage and monocytes owing to the small sample size [9]. Here, we used single-cell sequencing data from explanted lung tissue in a much larger dataset [10] to elucidate distinct macrophage and monocyte populations in the lungs of COPD patients that may play a relevant role in disease pathogenesis, including exacerbations and infections. Specifically, our primary aim was to discover the relevant subtypes of lung macrophages and monocytes in COPD [11]. We also determined whether this approach could lead to the discovery of novel therapeutic compounds for COPD and validated it using data from a clinical trial.

## 2. Methods

All statistical analyses were performed in R. Here are the details: (1) Version 4.1.1, Seurat, New York, NY, United States. (2) Version 1.0.0, Monocle 3, Seattle, WA, United States. (3) QIAGEN, Hilden, Germany. (4) Illumina, San Diego, CA, USA.

### 2.1. Quality Control and Cell Clustering

To investigate the subtypes of macrophages and monocytes and their roles in COPD, we analyzed a published single-cell sequencing (scRNA-seq) dataset obtained from Gene Expression Omnibus (GEO) (GSE136831 [10]). As the primary focus of our study was to elucidate the role of macrophage/monocyte populations in the pathogenesis of COPD, we restricted our analysis to alveolar macrophages, non-alveolar macrophages, classical (c) monocytes, and non-classical (nc) monocytes from 18 COPD and 28 control donor lungs, as defined in the original study [10]. After quality control, 7929 cells were retained for downstream analyses. We performed quality control and cell clustering with Seurat (version 4.0.5) [11]. We also filtered out cells with less than 200 or more than 2500 unique genes, cells with fewer than 2000 detectable genes and those containing more than 5% of mitochondrial genes.

A principal component analysis (PCA) [12] mapped the 1000 most variable genes (with the largest variance) into the top 20 principal components (PCs), enabling a reduction in the dimensionality of the data. We used a shared nearest neighbor (SNN) modularity optimization clustering method in Seurat to identify the relevant cell clusters based on these top PCs. To visualize the clustering results, we reduced the 20 PCs into two dimensions using uniform manifold approximation and projection (UMAP) [13].

### 2.2. Analysis of Differential Expressed Genes (DEGs)

We used MAST [14] to identify the differentially expressed genes (DEGs, or signature genes) between different clusters of interest. DEGs were defined as genes with an adjusted *p*-value of less than 0.05, accompanied by an absolute difference in the gene expression level (using log_2_-fold-change (|log_2_FC|)) that was larger than the threshold value of 0.5.

### 2.3. Pseudo-Time Analysis

We used Monocle 3 (version 1.0.0) [15,16] to perform a pseudo-time analysis. Ficolin (FCN)1-high classical monocytes, which was Cluster 2 in our analysis, was selected as the starting point for the trajectory analysis, as per a previous scRNA-seq study of lung-specific myeloid cells [17]. We then used Monocle to map out the evolutionary trajectory using a reverse graph embedding algorithm, superimposed the trajectory onto a UMAP plot, and assigned a pseudo-time value to each cell. We divided the pseudo-time series into intervals of 0.5 pseudo-seconds. For each interval, we calculated the proportionality of cells derived from COPD lungs (or control lungs) as well as the absolute number of cells belonging to each of the four immune cell types.

### 2.4. Ingenuity Pathway Analyses (IPA) and Gene Set Enrichment Analysis (GSEA)

Cluster-specific DEGs, in addition to their respective log_2_FC and adjusted *p* values, were uploaded to Ingenuity Pathway Analysis (IPA; QIAGEN, Hilden, Germany). An adjusted *p* < 0.05 threshold was used for all gene sets alongside a |log_2_FC| > 0.5 threshold for all comparisons. For each differentially expressed (DE) gene set, IPA generated *p*-values (threshold [–log (*p* value) > 1.3]); the predicted activation or inhibition status (positive or negative Z-scores) for representative canonical pathways, diseases, and biological functions; and upstream regulators, which were then used to identify the leading chemical and biological drug targets predicted to modulate the transcriptional signatures. The top pathways, as well as the disease and biological functions, were then plotted as shown in Figure 3. Full lists of the pathways, diseases, and functions, and the predicted chemical and biological drug targets can be found in Appendix A. Cluster-specific DEGs were also analyzed by Gene Set Enrichment Analysis (GSEA) using clusterProfiler 4.0 [18] and the Molecular Signatures Database [19,20].

### 2.5. Randomized Controlled Trial (RCT) Data to Validate the Effects of Fluticasone on the Expression of Specific Genes

One of the top therapeutic hits in IPA was fluticasone. To validate this predicted drug target in vivo, we used data from the DISARM trial, which was a RCT (clinicaltrials.gov NCT02833480) in which patients with COPD were treated for 3 months with (1) inhaled fluticasone (in combination with salmeterol), (2) inhaled budesonide (in combination with formoterol), or (3) inhaled formoterol alone [21]. Bronchoscopy was performed just prior to randomization when the patients had been free of inhaled corticosteroids for at least 4 weeks and then repeated 3 months later, following the completion of the treatment phase of the RCT. During the bronchoscopies, bronchoalveolar lavage (BAL) was performed in the right middle lobe or lingula. The details of the DISARM trial are provided elsewhere [21]. RNA-seq was performed on the BAL cell pellets using NovaSeq 6000 (Illumina, San Diego, CA, USA) at a sequencing depth of 55 million reads and were processed as previously described [22,23]; >80% of the BAL cells’ constituents were macrophages or monocytes, as previously described [24].

## 3. Results

### 3.1. Cell Clusters and Annotations

The distribution of the four cell types within the COPD/control samples is summarized in Table 1. Non-alveolar (airway or interstitial) macrophages were the most abundant cell population in the control lungs (76.48%), whereas alveolar macrophages were the most abundant of the evaluated cell types (34.27%) in the COPD lungs.

Cluster analyses identified 16 cell groups (Figure 1A). To investigate how these 16 clusters were related to COPD status, we calculated the proportion of cells belonging to each of the four immune cell types stratified by the lung origin of the cells (i.e., COPD versus control lung status; Figure 1B). We labeled the clusters as “COPD-predominant” if >70% of the cells in the cluster originated from the COPD lungs and as “control-predominant” if >70% of the cells in the cluster arose from the control lungs. The remaining clusters were labeled as “mixed” to reflect the heterogeneous origins of these cells. We also include Appendix A, which includes the purity of all 16 clusters.

We used the MAST [14] algorithm to annotate the discovered clusters and obtain the significant DEGs for each cluster relative to all the other clusters. The COPD-predominant clusters included alveolar macrophage Cluster 0 (high FABP4), non-classical monocyte Cluster 8 (high CD52), and classical monocyte Cluster 15 (high IL1B; Figure 1B). Control predominant clusters included seven non-alveolar macrophage groups: Cluster 1 (FABP4, CCL2, and IFI6), Cluster 3 (high MT1G), Cluster 4 (high SFTPC), Cluster 5 (high VSIG4 high), Cluster 7 (high SPP1 and S100A8/9), Cluster 9 (high BAG3), Cluster 13 (high SERPINB2), one classical monocyte Cluster 11 (high FCN1), and two alveolar macrophage Clusters 12 (high FABP4) and 14 (high MALAT1). Mixed clusters included classical monocyte Cluster 2 (high FCN1), non-classical monocyte Cluster 10 (high CD52), and non-alveolar macrophage Cluster 6 (high RGS1).

These clusters matched or were related to numerous previously reported myeloid clusters derived from human lung studies [17,25,26]. For example, *MT2A* and *MT1E* and *HMOX1* were identified as marker genes within the FABP4-high alveolar macrophage Clusters 0 (COPD-specific) versus 12 (control-specific) (Appendix A). These marker genes were also identified as markers of a COPD-specific alveolar macrophage cluster from a previous analysis of this dataset [25]. Marker genes for classical monocytes (*S100A12*, *VCAN*), non-classical monocytes (*LILRB2*, *LILRA5*), interstitial macrophages (*FPR3*, *MRC1*), and alveolar macrophages (*PPIC*, *AMIGO2*, *MRC1*) also matched this previous analysis [25] (Appendix A). Furthermore, marker genes for the MT1G-high non-alveolar macrophage Cluster 3 and the RGS1-high Cluster 6 matched those identified in BAL fluid of healthy controls and patients with COVID-19 [17] (Appendix A), while the SERPINB2-high Cluster 13 matched marker genes from BAL cells obtained in healthy and COPD lungs [26] (Appendix A).

### 3.2. A Pseudo-Time Analysis Revealed Three Significantly Different Cellular Evolutionary Trajectories

To determine how the cell types changed along the disease’s trajectory, we performed a pseudo-time analysis (trajectory inference, Figure 2). Similar to a previous study [17], we used the FCN1-high classical monocytes (Cluster 2) as the starting point for the inference of the trajectory. This pseudo-time analysis revealed three independent evolutionary trajectories (Figure 2). The first terminated at the non-classical monocyte clusters (Trajectory 1); the second ended at the alveolar macrophage clusters (Trajectory 2); and the third finished at the non-alveolar macrophage clusters (Trajectory 3). In Trajectory 1, a classical monocyte to non-classical monocyte transition occurred; while for Trajectories 2 and 3, the transitions were more complicated, with both pathways sharing a large number of transitional states before being directed to specific terminal genes. The control predominant clusters followed only Trajectory 3, while all COPD-predominant clusters followed either Trajectory 1 or 2. These trajectories are illustrated in Figure 2. A pseudo-time analysis including only the COPD samples confirmed similar responses, showing two trajectories, both starting from the classical monocyte clusters (Appendix A). Trajectory 1 ended at the non-classical monocyte clusters, while Trajectory 2 traversed through the non-alveolar macrophage clusters and terminated at the alveolar macrophage clusters. Trajectory 2 aligned with earlier findings, depicting a path from monocytes through non-alveolar macrophages en route to alveolar macrophages [27].

### 3.3. Differential Expression and Gene Set Enrichment Analysis of COPD-Predominant Clusters

To determine the gene expression signatures specific to COPD-predominant clusters, we performed a DE analysis (Appendix A) of COPD-predominant vs. control-predominant clusters. First, we investigated the FABP4-high alveolar macrophages in Cluster 0 versus the control-predominant FABP4-high alveolar macrophages in Cluster 12. This analysis revealed 176 DEGs (83 upregulated and 93 downregulated; P_adj_ < 0.05, |log_2_FC| > 0.5, Appendix A). According to IPA, these genes were enriched in downstream biological processes including poor lipid uptake and impaired activation, movement, and angiogenesis of cells (Figure 3A, Appendix A). These predicted (inhibitory) processes were accompanied by the inhibition of canonical pathways such as LXR/RXR activation (Figure 3A, Appendix A). Additionally, a GSEA of the upregulated genes demonstrated HALLMARK pathway enrichment of the interferon-γ response, IL6-JAK-STAT3 signaling, and apoptosis, while downregulated genes showed enrichment in adipogenesis, complement activation, and fatty acid metabolism (Appendix A).

Next, we investigated the COPD-predominant IL1B-high classical monocyte Cluster 15. Classical monocytes are considered to be highly phagocytic cells, with capacity for a potent inflammatory immune response and migratory functions [28]. Compared with the control-predominant classical monocyte Cluster 11, DE analysis identified 495 DEGs (297 upregulated and 198 downregulated; P_adj_ < 0.05, |log_2_FC| > 0.5, Appendix A). IPA of this gene set predicted the activation of numerous biological functions such as the activation, mobilization, and recruitment of cells, along with enrichment in carbohydrate metabolism, and increased respiratory bursts and inhibition of organismal death (Figure 3B, Appendix A). Enriched canonical pathways included activation of the cytokine/chemokine responses to influenza, pathogen-induced cytokine storm signaling, and TREM1 signaling (Figure 3B, Appendix A). These data suggest that classical monocytes specific to COPD lungs possess an activated and hyperinflammatory gene expression signature. GSEA confirmed these findings, with Gene Ontology (GO) enrichment of IL-1B and IL-6 production and increased neutrophil chemotaxis (Appendix A). When we compared the classical monocytes from COPD versus controls within the same cluster (mixed classical monocyte Cluster 2, n = 160), we found that relative to cells from the control lungs, those from the COPD lungs also demonstrated an activated and hyperinflammatory transcriptional state, alongside impaired phagocytotic function (Appendix A). A comparison of Cluster 15 with Cluster 2 (n = 374) also predicted a similar activated and hyperinflammatory gene signature.

### 3.4. Predicted Chemical and Biological Drug Targets in COPD-Predominant Monocyte and Alveolar Macrophage Clusters

We predicted the drug targets on the basis of the DEGs in the COPD-related clusters. To do this, we included an analysis of the differentially expressed genes from COPD-predominant vs. control-predominant clusters (e.g., alveolar macrophages (Cluster 0 vs. Cluster 12) and classical monocytes (Cluster 15 vs. Cluster 11), as well as mixed clusters (COPD vs. control cells within the same non-alveolar macrophage Cluster 6, classical monocyte Cluster 2, and non-classical monocyte Cluster 10). Interestingly, fluticasone propionate, a drug commonly used in COPD, was significantly enriched in all cluster comparisons, and was within the top five predicted drug targets across three of the five comparisons (Appendix A). Fluticasone propionate enrichment was driven by differentially expressed genes that are known to be regulated by this drug (references from IPA [29,30,31], incorporated within the IPA database; n = 47 across all comparisons). In four of the five comparisons (all except classical monocytes (Cluster 15 vs. Cluster 11)), fluticasone propionate was predicted to be inhibited. This suggests that administration of this drug would be predicted to reorient the gene expression of differentially expressed genes to those of control lungs. For Cluster 15 vs. 11, fluticasone was predicted to be activated, suggesting its administration would potentiate changes in gene expression. In vitro, fluticasone has been shown to have divergent transcriptional effects on monocyte-derived macrophages, weakening components of the adaptive immune signature while strengthening the chemokine and phagocytic attraction pathways [29].

### 3.5. Clinical Trial Evaluation

We validated the differential expression of fluticasone-regulated genes across monocyte and macrophage clusters identified by IPA in the DISARM trial [21]. The DISARM study’s design is shown in Figure 4A; 37 COPD subjects with BAL gene expression data from both before and after treatment were included in the current analysis. The baseline characteristics are shown in Appendix A. To validate the differential expression of fluticasone-regulated genes, we compared the pre- and post-treatment gene expression levels in the BAL cell pellets (which were >80% monocytes and macrophages; log_2_ counts per million [log_2_CPM]) within each treatment arm using Wilcoxon’s signed rank tests (Figure 4). Of the 47 fluticasone genes, 19.1% (nine genes: HLA-DPA1, FKBP5, HLA-DPB1, HLA-DRA, CD74, C1QB, LST1, PDK4, and HLA-DQA1) showed significant (*p* < 0.05) changes after salmeterol/fluticasone treatment, four genes (FKBP5, WARS, THBS1, and SLAMF1) significantly changed in the formoterol/budesonide group, and no genes showed significant changes in the formoterol-only group.

## 4. Discussion

Over the past 40 years, only one new class of therapeutics has been successfully introduced into the COPD market [32]. One important barrier has been the lack of an understanding of the key role that immune cells play in the pathogenesis of COPD. Although macrophages are the most abundant immune cells in the human airways and have been shown to play a crucial role in the pathogenesis of COPD [33,34], these cells have not been characterized well in the airway tissues of patients with COPD. Here, using single-cell sequencing technology, we showed that, transcriptionally, alveolar macrophages from COPD lungs demonstrate a dysfunctional phenotype, demonstrated by inhibition of the cell movement, angiogenesis, and lipid recycling pathways. Further, classical monocytes from COPD lungs displayed an activated hyperinflammatory gene signature compared with cells from control lungs, with activation of the hypercytokine and chemokine signaling and mobilization/recruitment pathways, alongside inhibited phagocytic functions. In silico analysis identified fluticasone as one of the top therapeutic compounds predicted to modulate the gene expression profiles of macrophages and monocytes from COPD lungs (and make them transcriptionally more similar to those from control lungs). We validated this finding in an RCT of BAL isolated from patients with COPD who were treated for 3 months with inhaled fluticasone. Together, these data highlight the power and promise of single-cell sequencing the macrophages and monocytes in lung tissue to identify new therapeutic targets and compounds in patients with COPD.

In this study, we identified 16 unique monocyte and macrophage clusters in lung tissue, extending previous studies utilizing single-cell RNA sequencing approaches in lung tissue and BAL from healthy and COPD patients [17,25,26,27]. Interestingly, 10 out of these 16 clusters predominantly comprised cells from control lungs, and 3 clusters were derived predominantly from COPD lungs. Among our 16 clusters, we identified the corresponding FABP4-high alveolar macrophage clusters, including COPD-predominant and control-predominant clusters. In the COPD lungs, the largest proportion of cells were alveolar macrophages and non-classical monocytes, whereas in the control lungs, non-alveolar (i.e., airway and interstitial) macrophages were by far the largest group. As expected, there were very few non-classical monocytes in the control lungs.

Our findings are in general agreement with those of previous studies. For example, we and others have shown previously that the macrophages isolated from COPD lungs demonstrate impaired phagocytosis, a key regulatory function of macrophages in vivo [35,36]. The perturbed phagocytic function of macrophages is thought to be one of the important contributors to impaired host defenses against inhaled pathogens in patients with COPD, as the macrophages in the lungs are largely responsible for clearing apoptotic cells and other debris in the airways (through efferocytosis [37]). A previous study has associated this phenotype with altered mitochondrial function and the inability to regulate the production of ROS [38]. Our work also aligns with the findings of Li et al., who performed scRNA sequencing of lung tissue (comparing severe COPD patients with controls) and showed that gene expression was most different (between COPD and control lungs) for monocytes and macrophages [39]. Here, we extend their findings by demonstrating the specific genes and pathways that are altered in COPD and by identifying 16 different clusters of macrophage/monocyte subpopulations in human lungs. We showed that 3 months of treatment with fluticasone but not budesonide could significantly modulate gene expression of BAL cells (which are mostly macrophages/monocytes). These data suggest that gene responsiveness of macrophages/monocytes in COPD may be compound-specific. Additional studies will be required to validate this notion.

There were some limitations to this study. First, we restricted our evaluation to macrophage/monocyte populations in lung tissue and did not consider the transcriptomic profile of other immune cells such as neutrophils, eosinophils, dendritic cells, or lymphocytes, which may also play a significant role in the pathogenesis of COPD. We undertook this study because macrophages, being the most abundant immune cells in the lungs, have a pivotal role in the pathogenesis of COPD; however, their precise contribution remains incompletely understood. Furthermore, it is noteworthy that they have not constituted the primary focus of drug discovery efforts in COPD [8]. Second, owing to the relatively small sample size, we did not investigate macrophage/monocyte subpopulations across the full spectrum of disease severity of COPD. Third, the pathogenesis of COPD is complicated and may involve a complex set of interactions among immune, neural, vascular, and structural cells. Nevertheless, given the importance of monocytes and macrophages in lung homeostasis and maintenance, these cells are likely to play a pivotal role in the disease’s pathogenesis. Further studies are needed to discern how dysfunction of the myeloid cell contributes to inflammatory dysregulation and/or cell dysfunction in other lung-specific cell types to drive pathology of COPD. Fourth, we did not validate the transcriptional signatures with functional studies. However, we have shown previously that over 50% of macrophages from COPD airways cannot be phenotyped using traditional M1/M2 cell surface markers (unlike macrophages from control airways, in which >80% of the cells are typeable), exhibit mitochondrial gene defects, and demonstrate impaired motion and phagocytosis ex vivo [24,36,40]. Finally, a previous study used RNA velocity analysis to map the cellular transitions within alveolar macrophage clusters [26]. Owing to the limited availability of the original data, we were unable to apply an RNA velocity analysis to our dataset; thus, we could not recapitulate the cellular transitions within our alveolar macrophage clusters.

Notwithstanding these and other limitations, our study findings have important clinical implications. First, COPD lungs harbor macrophages and monocytes that are predicted to have poor homeostatic and antimicrobial properties, which may explain the increased risk of exacerbation, pneumonia, and other infectious complications in COPD patients. Second, the transcriptional profile of these cells can be altered with therapeutic drugs such as fluticasone, which may also extend to other identified compounds such as ruxolitnib and decitabine (Appendix A). These latter compounds should be investigated for their therapeutic potential in patients with COPD. Third, COPD lungs contain pro-inflammatory macrophage/monocyte populations, which may explain the heightened inflammatory status of COPD lungs, even in the absence of an active infection, and suggest that targeted anti-inflammatories may be effective in improving the health status and outcomes of COPD patients.

## Figures and Tables

**Figure 1 cells-12-02771-f001:**
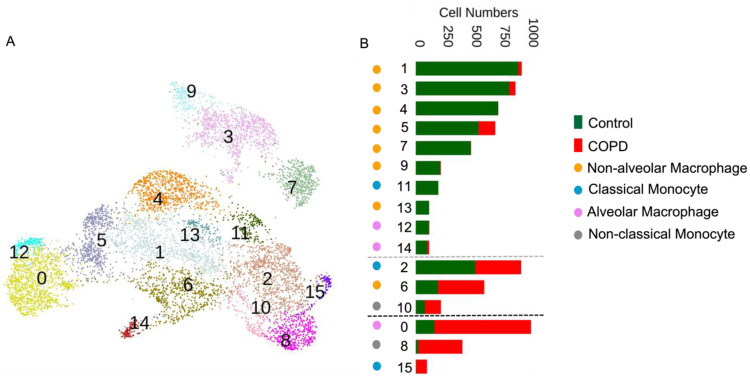
(**A**) UMAP plot for all cell clusters. (**B**) The bar chart depicts the number of immune cells according to the COPD or control status for each cluster. Clusters are displayed according to the distribution of the four immune cell types. Clusters are shown according to the origins of the cells (COPD or control lungs). A cluster was deemed to be “COPD-predominant” if more than 70% of the cells originated from the COPD lungs (i.e., clusters below the black horizontal dashed line) and “control-predominant” if more than 70% of the cells originated from control lungs (i.e., clusters above the grey horizontal dashed line). The remaining clusters were deemed to be mixed in terms of their origin and are displayed as bars between the two horizontal dashed lines. The bars in each subcategory are arranged in descending order of their respective cell numbers from left to right.

**Figure 2 cells-12-02771-f002:**
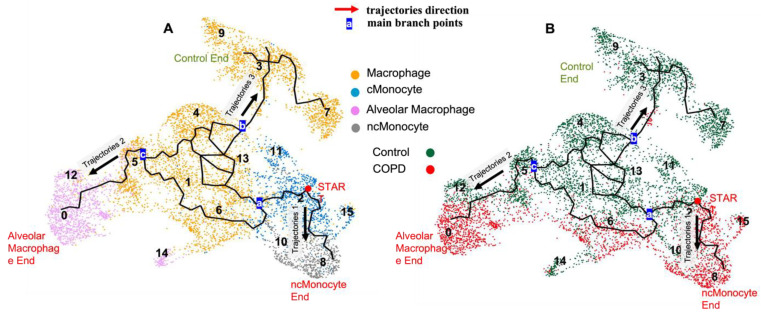
Pseudo-time trajectories projected on a UMAP graph. (**A**) Cells are colored according to the cell types. Non-classical (nc) monocytes are colored grey in Clusters 8 and 10; classical (c) monocytes are colored sky blue in Clusters 2, 11, and 15; macrophages are colored orange in Clusters 1, 4, 5, 6, and 13 in the middle area and Clusters 3, 7, and 9 in the upper area; and alveolar macrophages are colored pink in Clusters 0, 12, and 14. (**B**) Cells colored according to the disease status (i.e., COPD/control). Control cells are in green, and COPD cells are in red. Three different trajectories were inferred. The green and red arrows indicate the directions of the three different trajectories. The labels a, b, and c represent three main branching points in these trajectories.

**Figure 3 cells-12-02771-f003:**
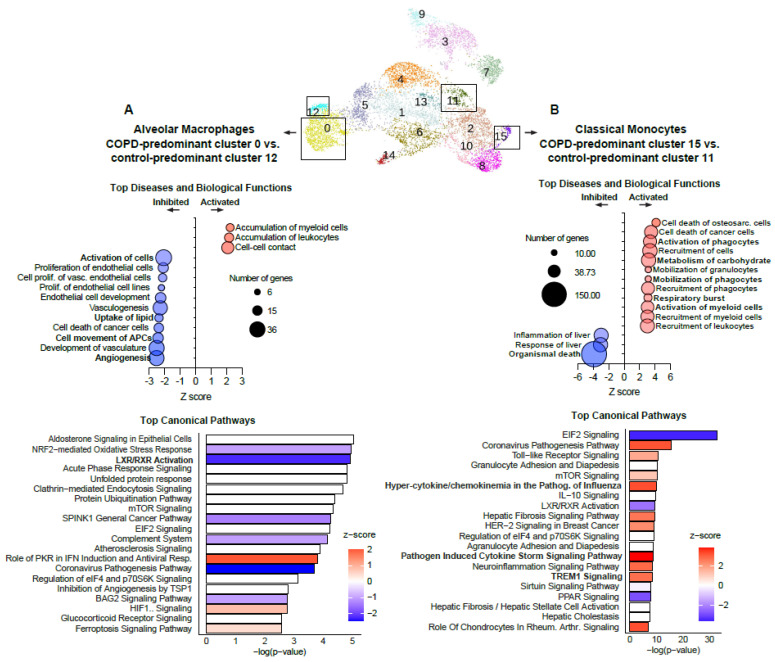
Top downstream diseases and biological functions and canonical pathways from COPD-predominant vs. control-predominant clusters predicted using Ingenuity Pathway Analysis. (**A**) Downstream diseases and biological functions (top) and canonical pathways (bottom) of differentially expressed genes in FCN1-high alveolar macrophages from the COPD-predominant Cluster 0 vs. the control-predominant Cluster 12 (n = 176, Padj < 0.05, |log2FC| > 0.5). (**B**) Downstream diseases and biological functions (top) and canonical pathways (bottom) of differentially expressed genes in classical monocytes from the COPD-predominant Cluster 15 vs. the control-predominant Cluster 11 (n = 495, Padj < 0.05, |log2FC| > 0.5).

**Figure 4 cells-12-02771-f004:**
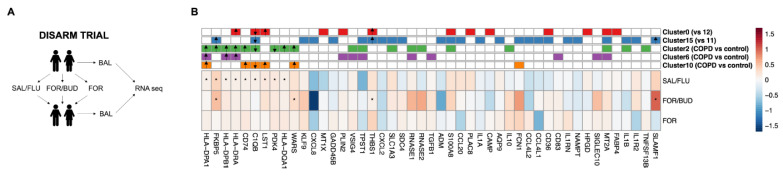
Validation of the IPA-predicted fluticasone genes in DISARM. (**A**) A graphical summary of the DISARM trial. Patients with stable COPD underwent bronchoscopy and a BAL procedure before treatment, with the patients having been free from inhaled corticosteroids for at least 4 weeks. One week later, the patients were randomized to three arms and treated with (1) salmeterol/fluticasone (SAL/FLU), (2) formoterol/budesonide (FOR/BUD), or (3) FOR for 3 months. Another BAL sample was taken at the end of the treatment period. (**B**) Heat map of the changes in gene expression (median post-treatment minus pre-treatment log2 counts per million (Δlog2CPM)) in each treatment group. * represents Wilcoxon’s signed rank test *p*-value < 0.05, and color represents the median Δlog2CPM. Genes were arranged according to their significance in the SAL/FLU group. The top rows contain annotated genes that were predicted to be regulated by fluticasone by IPA from each single-cell comparison.

**Table 1 cells-12-02771-t001:** Distribution of monocytes/macrophages in COPD versus control lungs, with row percentages in brackets.

	Macrophages	Alveolar Macrophages	cMonocyte	ncMonocyte	Total
COPD	666 (25.67%)	889 (34.27%)	506 (19.51%)	533 (20.55%)	2594
Control	4080 (76.48%)	455 (8.53%)	651 (12.20%)	149 (2.79%)	5335
Total	4746 (59.85%)	1344 (16.95%)	1157 (14.59%)	682 (8.61%)	7929

Abbreviations: COPD, chronic obstructive pulmonary disease; cMonocyte, classical monocyte; ncMonocyte, non-classical monocyte.

## Data Availability

There is no new data were created.

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
