# Peer review of "Single-Cell Sequencing of Lung Macrophages and Monocytes Reveals Novel Therapeutic Targets in COPD"

_cells, 2023, doi:10.3390/cells12242771_

Round 1

Reviewer 1 Report

Comments and Suggestions for Authors

This is an interesting paper that studies single-cell sequences of lung macrophages and monocytes in COPD lungs.  The paper describes alterations in gene structure in these cells.

The paper could be improved if the findings in this study could more directly be related to the major pathological mechanism, such as lung elastin degradation and airway inflammation. I suggest requesting the authors to revise the paper to address these concerns.

Reviewer 2 Report

Comments and Suggestions for Authors

He et al. report an analysis of a publicly available scRNA-Seq dataset that comprises both COPD, IPF and control lung cell samples.  In their analysis, He et al. aim to understand the pathological features and dysfunctions of monocytes and macrophages in COPD, thereby expanding beyond the original analysis reported by Adams et al.

There are a few reports that should be referenced and discussed, as these are very similar to those reported in this paper.  The first one is a paper by Sauler et al., in which scRNA-Seq was used to determine the phenotype of alveolar macrophages in COPD (doi: 10.1038/s41467-022-28062-9).  Adams et al. study (published in 2020) was performed in Kaminski lab (Yale University) who further characterized COPD lungs tissues by scRNAseq in 2022 as reported in Sauler et al. (doi: 10.1038/s41467-022-28062-9) and uncovered a new subset of dysregulated alveolar macrophage expressing metallothioneins (MT1G, MT2A) in COPD lungs. Similarly to Mould’s transcriptomic analysis in healthy human BALs (doi: 10.1164/rccm.202005-1989OC), Liegeois et al profiled by single-cell RNA sequencing (doi: 10.1165/rcmb.2021-0563OC) distinct subsets of AMs, monocyte-derived AMs and monocytes in COPD BALs.  The pre-print by Ayaub et al. (doi: 10.1101/2021.01.04.425268) uses the same dataset as this publication by He et al., but focuses on the analysis of macrophages in idiopathic pulmonary fibrosis.  As much is shared between this report and the ones above, the authors should mention these and discuss their data relative to these other relevant reports.  It is also an over statement that it is novel to “use a modern bioinformatics tool, which harnessed transcriptional profiles at the resolution of single cell sequencing, to identify distinct clusters of macrophage/monocyte populations in the lung tissue” given the existence of these reports, and the statement should be revised. 

In the current manuscript, the authors clustered, annotated, and classified their monocyte/macrophages into four cell types: classical monocytes, non-classical monocytes, alveolar macrophages and non-alveolar macrophages. This classification implies that monocyte-derived macrophages, which are also thought to contribute to COPD pathology, may be included in the non-alveolar macrophages cluster. In addition, two distinct subsets expressing metallothioneins have been depicted: MT1G high macrophages in control-predominant cluster 3 and COPD-predominant cluster 12 which is enriched with MT2A and MT1E (Supp FigS3). The reader may question how clusters 3 and 12  relate to the dysregulated alveolar macrophage expressing metallothioneins (MT1G, MT2A) in COPD lungs described in the Kaminski study?

To provide better clarification of the cluster’s identities, it is recommended to:

1-     annotate the 16 identified clusters by their transcriptional signatures using a dot plot or heatmap showing the expression of marker genes across cell clusters, including cluster 10 (which is not described in the manuscript) and  cluster 4 (SFTPC high macrophage that dominate in control samples), and discuss the relevance of this clustering to Kaminski’s and Liegeois’ studies.

2-     highlight whether non-alveolar macrophages include monocyte-derived subsets by verifying the expression of monocyte and macrophage markers in theses subsets

The authors report a pseudo-time analysis, which infers lineage relationships based upon similarities in gene expression.  The analysis was done on the combined data from COPD and control subjects.  This is valid if the same lineage trajectories hold under normal vs. chronic inflammatory conditions.  The analysis should be supplemented with one done on COPD samples only to validate that the predictions hold true under chronic inflammatory conditions.

The authors reported significant differences in classical monocytes between COPD and control subjects, with data suggesting a hyperinflammatory gene expression signature of COPD-predominant cluster 15 compared to the control-predominant cluster 11.  Between these who clusters sits one with roughly equal contributions from COPD and controls, cluster 2.  It would be interesting to see a comparison of the COPD cluster (15) to this mixed cluster (2) discussed to understand how this COPD-specific state compares to one that is also observed in both health and disease.

To determine disease-specific mono/macrophages subtypes, the authors performed trajectory interference analysis by considering classical monocytes (Cluster 2) as origin of on all others. This analysis terminated at COPD alveolar macrophages (Cluster 12) that could transition through clusters 6 and 5, and COPD non-classical monocytes (Cluster 8). However, Liegeois’s work identified distinct subsets of macrophages including altered AMs, classical AMs, self-proliferative AMs and monocyte-derived macrophages where RNA velocity analysis (another method for evolutionary trajectory analysis) was applied to confirm that altered COPD AMs shifts from classical AMs rather than deriving from monocyte-derived macrophages.

The authors are invited to comment on, how COPD AMs (Cluster 0) relates to Liegois AM clusters? whether Cluster 0  is heterogeneous bearing self-proliferative macrophages? Whether AMs in cluster 0 are altered and shifting from Cluster 12 or they derive from classical monocyte (cluster 2) by transitioning through clusters 5 and 6, which implies that these clusters may represent monocyte-derived macrophages? In this case, how do clusters 5 and 6 fit into Liegeois clustering of monocyte-derived macrophages subsets (pro-inflamed, wound healing and chemokine-secreting subsets) that were found impaired in COPD BALs?

In their drug predictions to restore COPD monocyte and macrophage state, the authors use Ingenuity Pathway Analysis, identifying enrichment of signals related to fluticasone propionate in all their comparisons.  However, when looking at the supplementary table, the directionality is not consistent between the comparisons.  In particular, the comparison of clusters 15 to 11 identified previously as a potentially hyperinflammatory phenotype of classical monocytes in COPD shows activation of the fluticasone signature, while the other comparisons show the opposite.  The authors should report the directionality in their results section (it is only mentioned for the first time in the discussion, and not taking the differing directionalities into consideration).  It would be interesting to hear the authors’ take on why they see activation in cluster 15 compared to 11?  Is cluster 15 a drug-induced effect, a disease-driver, or both?  The authors can consider further discussing the paper by van de Garde et al. which they cited, as it investigated fluticasone propionate effects on monocyte-derived macrophages in vitro as well as using a cigarette smoke mouse model.

To validate the fluticasone finding, the authors used Bulk RNA seq data from the BAL samples included in the DISARM study, which is a nice approach.  Given that sc-RNAseq of monocytes/macrophages in human BALs were made publicly available by Mould et al (doi: 10.1164/rccm.202005-1989OC) in healthy donors and by Liegeois et al (doi: 10.1165/rcmb.2021-0563OC) in COPD BALs, the authors are invited to:

1-     Justify the rationale behind choosing COPD lungs for mono/macrophages clustering instead of BALs, specifically that single cell RNAseq data from COPD BALs is publicly available.

2-     Comment on the novelty of their trajectory interference approach in identifying the evolutionary paths toward altered COPD monocytes/macrophages by compared it to Mould’s and Liegeois's studies.

In addition, their results are  difficult to evaluate as letters in figure 4B have been replaced by characters (at least when I view it).  If this is generally true it must be fixed.  Also, the top rows that indicate the genes predicted to be regulated by fluticasone in the respective clusters should also show  the directionality in which they are dysregulated in COPD compared to control, e.g. by up/down arrows.  I have to defer any evaluation of these results until the data presentation has been corrected.

Fluticasone is predicted to restore the gene expression profiles of COPD macrophages/monocytes toward healthy states. Given the insufficient evidence highlighting the changes in frequencies, transcriptional and functional profiles in each cluster before and after fluticasone treatment in COPD patients, the authors seem to overinterpret their data by stating that their findings were validated in the COPD patients.

In addition to this, there are some minor comments:

·        A heat map showing the cell types analyzed (alveolar macrophages, non-alveolar macrophages, classical and non-classical monocytes) and their expression of marker genes should be provided as a supplementary figure to clarify the cell classifications

·        As a supplementary figure, please include a similar UMAP plot and bar graph as fig. A but identifying the distribution of cells from each subject and not only disease state. Are any of the clusters dominated by one or a few subjects, or are these general observations across samples?

·        The 16 "distinct clusters" still only remain phenotypical observations by gene expression.  It remains unknown if these comprise functionally distinct subsets of cells.  This should be discussed in the limitations of this study.

·        Authors of reference 28 have been omitted.

·        Incorrect references in the discussion: ref. 8 is not by Li et al., ref. 37 is not by Chan et al.  Please correct.

Comments on the Quality of English Language

Fair

Round 2

Reviewer 2 Report

Comments and Suggestions for Authors

Dear authors

thanks for responding to the comments

my responses are below

C1.  Addressed.

C2.  Mostly addressed.  Some clarifications required:

  1. Include a reference to supplementary table S9 in the main text.  Also, are the Sftpc-high macrophages (cluster) enriched for any specific markers?  Now they according to table S9 appear as a mixture of CCL2-high, CCL18-high, and MT1G-high cells.  Is the Sftpc derived from the macrophages themselves, or through processes of AT2 cell efferocytosis?
  2. Is HMOX1 a marker gene for the FABP4-high cluster?  The volcano plot suggests that it's in the opposite direction to MT2A and MT1E.
  3. While cluster 3 resembles the MT1G-high cluster in Wauters et al., the RGS1-high cluster does not seem to match up very well with any specific cluster in this paper.
  4. I don't see a clear match between cluster 13 and any of the clusters from Liegeios et al.

C3. Addressed.

C4. Addressed.

C5. Since the current study is not the first to identify different COPD-specific mono/macrophages and to determine evolutionary paths toward altered COPD monocytes/macrophages, the authors should comment in the discussion section the novelty of their clustering and discuss the novelty of their trajectory interference compared to what is already published.

C6. Addressed.

C7. Similar to comment 5.

C8. Addressed.

C9.

  1. Addressed, but table S9 should be referenced in the main text as previously noted.
  2. Not addressed.  Should at least be commented on.
  3. Addressed.
  4. Addressed.
  5. Chana et al. has been replaced by Kapellos et al [8] in the statement.  This is a review article.  Was this the primary article meant to be cited?
